# Plant Viruses Can Alter Aphid-Triggered Calcium Elevations in Infected Leaves

**DOI:** 10.3390/cells10123534

**Published:** 2021-12-14

**Authors:** Christiane Then, Fanny Bellegarde, Geoffrey Schivre, Alexandre Martinière, Jean-Luc Macia, Tou Cheu Xiong, Martin Drucker

**Affiliations:** 1PHIM, INRAE, CIRAD, IRD, Institut Agro, University Montpellier, 34980 Montferrier sur Lez, France; nanethen@web.de (C.T.); jean-luc.macia@inrae.fr (J.-L.M.); 2BPMP, University Montpellier, CNRS, INRAE, Institut Agro, 34060 Montpellier, France; fannyb@nuagr1.agr.nagoya-u.ac.jp (F.B.); geoffrey.schivre@gmail.com (G.S.); alexandre.martiniere@cnrs.fr (A.M.); 3Graduate School of Bioagricultural Sciences, Nagoya University, Chikusa, Nagoya 464-8601, Japan; 4SVQV, UMR1131, INRAE, Université Strasbourg, 68000 Colmar, France

**Keywords:** *Arabidopsis thaliana*, cauliflower mosaic virus, turnip mosaic virus, turnip yellows virus, green peach aphid, aphid feeding activity, calcium signalling, defences, transmission

## Abstract

Alighting aphids probe a new host plant by intracellular test punctures for suitability. These induce immediate calcium signals that emanate from the punctured sites and might be the first step in plant recognition of aphid feeding and the subsequent elicitation of plant defence responses. Calcium is also involved in the transmission of non-persistent plant viruses that are acquired by aphids during test punctures. Therefore, we wanted to determine whether viral infection alters calcium signalling. For this, calcium signals triggered by aphids were imaged on transgenic *Arabidopsis* plants expressing the cytosolic FRET-based calcium reporter YC3.6-NES and infected with the non-persistent viruses cauliflower mosaic (CaMV) and turnip mosaic (TuMV), or the persistent virus, turnip yellows (TuYV). Aphids were placed on infected leaves and calcium elevations were recorded by time-lapse fluorescence microscopy. Calcium signal velocities were significantly slower in plants infected with CaMV or TuMV and signal areas were smaller in CaMV-infected plants. Transmission tests using CaMV-infected *Arabidopsis* mutants impaired in pathogen perception or in the generation of calcium signals revealed no differences in transmission efficiency. A transcriptomic meta-analysis indicated significant changes in expression of receptor-like kinases in the BAK1 pathway as well as of calcium channels in CaMV- and TuMV-infected plants. Taken together, infection with CaMV and TuMV, but not with TuYV, impacts aphid-induced calcium signalling. This suggests that viruses can modify plant responses to aphids from the very first vector/host contact.

## 1. Introduction

Plants sense their environment and respond to abiotic and biotic cues and stresses by installing defences. This includes responses against infestation by phloem feeders such as aphids. Plant defences against aphids seem to follow classical pathogen-associated molecular pattern-induced pathways (for example, [1,2]), analogous to responses against fungi and bacteria [3], but the genes and effectors involved remain largely unknown [4]. Aphids have a particular feeding behaviour [5,6]. After having landed on a potential new host plant, they first insert their needle-like stylets into the tissue. During the advancement of the stylets between the cells, they secrete gelling saliva that forms a sheath that constitutes a tight seal around the stylets, which insulates them from the tissue. By puncturing epidermis and mesophyll cells, aphids test the plant for suitability and locate the phloem. Aphids first secrete watery saliva into a punctured cell and then aspire some of the cell contents to probe it. Then the stylets are retracted and probing proceeds further into the plant tissue until they reach the sieve tubes. The feeding behaviour then changes. The stylets remain inserted into the tapped sieve tube and brief salivation phases alternate with long suction phases during which phloem sap, the principal food source for aphids, is acquired. Gelling saliva and especially saliva components injected into either parenchyma cells or into the sieve tubes are assumed to contain aphid effectors that control aphid–plant interactions and can be recognised by the plant just like effectors from other pathogens (for recent examples, see [7,8]; for review, see [4]). A very early step in perception and response to biotic stress is the fast increase of free calcium in the cytosol, induced by pattern recognition receptors (for review, see [9,10]). Recent work showed that aphid punctures trigger calcium elevations around feeding sites [11], suggesting a link between calcium signalling, plant perception of aphids, and subsequent local and systemic defence reactions. Meta-analysis of transcriptomes also indicates a role for calcium signalling in aphid–plant interactions [12].

Many plant viruses are transmitted by arthropods such as aphids, which acquire viruses while feeding on infected plants. Two basic transmission modes are discerned: circulative and non-circulative (for review, see [13]). In circulative transmission, food-contained virus particles (virions) traverse the intestine of the vector, circulate through the hemolymph, invade the salivary glands, and are inoculated with the saliva into new host plants. In non-circulative transmission, however, virions are retained in and released from the external mouthparts of the vectors (the stylets and/or the foregut) to a new host. It was shown that the acquisition of two aphid-transmitted non-circulative viruses was not the accidental contamination of vector mouthparts during feeding but was controlled by specific virus–vector–plant interactions: cauliflower mosaic virus (CaMV, family *Caulimoviridae*) and turnip mosaic virus (TuMV, family *Potyviridae*) react within seconds to the presence of aphid vectors on infected plants and form specific transmission morphs that are efficiently acquired and transmitted [14,15,16,17]. Such responses require perception and signalling events whereby the plant recognises aphid feeding and initiates defence responses, which might be intercepted by the viruses. Indeed, lanthanum, a general calcium channel blocker, inhibits the transmission of CaMV and TuMV [15]. Hence, there is evidence that calcium signalling participates in the transmission of two non-circulative viruses. To our knowledge, no equivalent data are available for the transmission of circulative viruses. However, it was reported that vector-induced early calcium elevations could trigger antiviral plant defences [18].

Since the modulation of calcium signalling has been reported for some human viruses (for example, see [19,20]) and because calcium signalling is involved in both aphid–plant interactions and in the acquisition of CaMV as well as TuMV, we were interested to know whether the viral infection alters calcium signals. For this, we used a transgenic *Arabidopsis* line expressing the cytosolic calcium reporter YC3.6-NES [21]. YC3.6 is a well-characterized calcium sensor allowing the sensitive real-time measurement of free calcium levels in cells and tissues [22,23]. YC3.6-NES plants were infected with the non-circulative viruses CaMV and TuMV. The circulative turnip yellows virus (TuYV, family *Luteoviridae*) was also tested because no information on calcium interference of circulative viruses is available. Calcium signals triggered by aphids on healthy plants or infected plants were monitored by epifluorescence microscopy and analysed in detail.

## 2. Materials and Methods

### 2.1. Aphids, Plants, and Viruses

Wingless *Myzus persicae* aphids were maintained on eggplants (*Solanum melongena* cv. Barbentane) under controlled conditions (22/18 °C day/night with a photoperiod of 14/10 h day/night) in insect-proof cages.

The transgenic *Arabidopsis thaliana* line expressing the cytosolic calcium reporter YC3.6-NES under control of the constitutive ubiquitin 10 promoter [21] was grown under controlled conditions at 20/17 °C day/night and 60% relative humidity with an 8/16 h day/night photoperiod. Seeds were planted with Humin-substrat N2, pH 5.8 (Neuhaus, Geeste, Germany) and watered with a nutrient solution (N 150 mg/L, P 100 mg/L, K 300 mg/L, CaO 150 mg/L, MgO 40 mg/L) until transfer to the imaging platform where they were watered with tap water.

*Arabidopsis thaliana* mutants used in transmission experiments are listed in Table 1.

CaMV strain Cabb B-JI [24] and TuMV strain C42J [25] were mechanically inoculated into 4-week-old *Arabidopsis* YC3.6-NES plants using infected leaves crushed in water as inoculum and carborundum to facilitate penetration. The recombinant TuYV mutant TuM1s81 [26], containing an insertion for silencing the *AtCHLI1* gene, which facilitates the identification of infected plants by bleaching veins, was initially agroinfiltrated into 4-week-old *Arabidopsis* YC3.6-NES plants, and then propagated by aphids to new plants.

Infected plants were used for experiments when symptoms were clearly visible (3–6 weeks after inoculation).

### 2.2. Image Acquisition

A detached leaf was transferred into a 30 mm Petri dish filled with water and incubated for 30–60 min to allow recovery from the wounding stress caused by leaf detachment [27]. Care was taken to use leaves showing systemic symptoms but no necrotic lesions and of comparable age for infected and healthy control leaves, i.e., the 3rd to 5th youngest leaf. Only non-viruliferous aphids were used for the experiments. They were starved for at least one hour in a cell culture flask that was humidified with moistened absorbent paper. Then, a single aphid was positioned with a small paint brush on the adaxial epidermis and image acquisition was started when the aphid stopped moving and positioned the antennae on its back, which is an indication that it had started feeding behaviour. If no calcium elevations were observed or if the aphids were obstreperous for more than 30 min, the recording was stopped, and a new aphid was placed on the leaf. Leaves were regularly changed. The inverted microscope used was a Zeiss Axiovert 200M (Zeiss, Jena, Germany) equipped with a 5× objective (Plan-Neofluar, NA = 0.16), and a Spectra 7 (Lumencor, Beaverton, OR, USA) light source for excitation at 438 nm (BP24). Light emission at 480 ± 20 and 535 ± 15 nm was collected thanks to a filter wheel (Sutter 10B). Fluorescence and bright-field images were captured with a CoolSnap HQ camera (Roper, Sarasota, FL, USA) every 5 s for 10 to 50 min and the focus was adjusted when required.

### 2.3. Image Analysis

Calcium signals were analysed with Fiji (NIH, Bethesda, MD, USA) and MATLAB R2016b (MathWorks, Natick, MA, USA). Dedicated scripts were written (available upon request) for automatizing the analysis. Briefly, calcium signals were first assessed with the ratio of FRET (535 nm) and CFP (480 nm) channels. Pixels of the ratio images were grouped by blocks (26 × 29 pixels) and the mean signal in each block was compared to the median signal of the whole image to allow for the detection of calcium signals. The start and end times of the calcium signal were identified from the time series of the block mean signal. Derivative images were then computed to trace the calcium variations. The calcium peak and amplitude were then quantified on each calcium signal. Pixel values of derivative images over the time were plotted and signal integration between the start and end time was estimated to obtain the peak values of the calcium signal. The standard deviation projection of derivative images from the beginning to the end of the calcium signal yielded the area of calcium signals. The circularity factors of the signal area were then determined. The calcium signal speed was estimated for 16 directions, taking the origin of the signal as the starting point. The speed in each direction was calculated by tracking the calcium signal front from the derivative images. For each direction, the highest speed was indicated in the figure. The average speeds were deduced from the 16 directional speeds.

### 2.4. Transcriptome Analysis

Transcriptome data from TuMV-infected (13 dpi) vs. mock-inoculated *Arabidopsis* Col-0 plants were from the experiment performed by [28], available in GEO (accession number GSE46760). The data from the 4 control and 3 TuYV-infected replicates were analysed with GEO2R (an R script implementation on GEO based on Biobase 2.30.0, GEOquery 2.40.0, and limma 3.26.8) using standard settings. The GB accession numbers of the output were converted to TAIR IDs using db3db on https://biodbnet-abcc.ncifcrf.gov (last accessed on 4 March 2021).

Transcriptome data from CaMV-infected (21 dpi) vs. mock-inoculated Col-0 plants were from the unpublished experiment performed by Voinnet et al. and deposited in GEO (accession number GSE36457). Processed data were downloaded from http://urgv.evry.inra.fr/cgi-bin/projects/CATdb/cons_diff.pl?project_id=100&experiment_id=152 (last accessed on 10 March 2021).

Only genes with a more than twofold change in expression and an adjusted *p*-value of <0.05 were considered for analysis. A list of candidate genes (GLR, calcium-related proteins, receptor-like kinases), was compiled and compared with significantly deregulated genes using the Join two Datasets tool on Galaxy (https://usegalaxy.eu/, last accessed on 24 March 2021).

### 2.5. CaMV Transmission Tests

Wingless *M. persicae* aphids were starved for 1 h, then placed on *Arabidopsis* wild-type (Col-0) or mutant plants that had been infected mechanically with CaMV Cabb B-JI 3 weeks before when the plants were 2 weeks old. For most tests, individual aphids were transferred after a 1 min acquisition access period to 10-day-old (cotyledon stage) turnip plants (*Brassica rapa* cv. “Just Right”) for virus inoculation. Twenty to 48 test plants were inoculated per condition. After 1 h inoculation, the aphids were killed with 0.2% Pirimor G (CERTIS Europe, Guyancourt, France). The test plants were transferred to a climate chamber and cultivated at 23/15 °C day/night and with a 12/12 h day/night photoperiod. Infected plants were identified 3 weeks later by visual inspection.

### 2.6. Figures and Statistical Analysis

Figure charts were generated with Prism 9 (GraphPad, San Diego, CA, USA). Statistic tests (Mann–Whitney test, *t*-test, and one-way ANOVA) were performed with Prism 9. We used Fisher’s exact test for the analysis of transmission tests with 2 nominal variables (infection status and plant mutant) and tests were run using the R implementation on https://biostatgv.sentiweb.fr/&gt;?module=tests/fisher (last accessed on 9 April 2021).

## 3. Results

### 3.1. Aphids Trigger Ring-Like Calcium Waves

Time lapse fluorescence microscopy was used to measure calcium elevations triggered by aphid punctures. For this, one aphid at a time was placed on a healthy or infected YC3.6-NES leaf. Time lapse recording was started once the aphid became still, suggesting that it had settled and started probing, the first step in feeding behaviour. Of all the recordings, 57 elevations of free calcium on healthy plants, 32 on CaMV-infected, 46 on TuMV-infected, and 55 elevations on TuYV-infected plants were selected for analysis; the other recordings were discarded because they were out of focus, the puncture was not recorded completely, or the leaf moved too much during recording. In most cases, the calcium signals propagated in all directions on the leaf surface, showed circular patterns, and remained restricted to the aphid-infested region of the leaf (Figure 1). The calcium waves originated from a point-like centre and covered a circular area in the early phase (up to 10 s). Then the propagation continued as a ring-like front, while the inner zone returned to basal calcium concentrations. With further extension, the excitation front became weaker and disappeared, meaning that the signal had collapsed. The average duration of calcium signals was around 30 s (Appendix A). Image frames of typical calcium recordings in non-infected and virus-infected leaves are shown in Figure 1. Videos showing calcium elevations are presented in Appendix A.

### 3.2. Characterisation of the Calcium Waves

We analysed the calcium signal propagations in detail to see whether they were modified by viral infection. First the calcium signal peaks (sum of signals over time) were quantified. Representative calcium signal peaks triggered by aphid punctures in healthy control leaves or leaves infected with the viruses are shown in Figure 2A. Signal peaks obtained from all the conditions were variable (Figure 2B). No significant differences of the calcium signal peaks between the different conditions were found (*p* > 0.05, one-way ANOVA test). Then, the duration and the amplitudes of the calcium signals were compared and likewise revealed no difference between the control and infected plants (Appendix A).

Next, the maximum intensity projections of calcium signals were compared between the control and infected leaves (Figure 3). Again, the circular nature of the signal is clearly visible. Therefore, the circularity factor of the calcium signals was estimated for each condition. No significant differences between the healthy control and infected leaves were found for the circularity factor (Appendix A). However, when the areas of calcium maximum projections were measured, significant differences were detected. In healthy control leaves, the average area was 4.09 × 10^4^ ± 0.04 × 10^4^ µm^2^. In the leaves infected with CaMV, TuMV, or TuYV, the average areas were 3.0 × 10^4^ ± 0.34 × 10^4^, 3.62 × 10^4^ ± 0.42 × 10^4^ and 3.95 × 10^4^ ± 0.48 × 10^4^ µm^2^, respectively (Figure 3B). Thus, the plants infected with CaMV showed a significantly smaller area of calcium signals.

We wanted to know whether the speed of calcium propagations was similar in all directions. For this, the velocity of the calcium wave was measured in all directions and plotted as a function of the angle. Figure 4A shows the directional propagation of representative calcium signals. The speed of calcium signals was slightly different in each direction, starting from the origin of the signal. The differences between minimum and maximum speeds (red and green arrows, respectively) were between 1.5 and 2 µm/s, meaning that the velocity varied by about 30–40%, depending on the direction of propagation. For each calcium wave, we averaged the directional velocities (as presented in Figure 4A) in order to obtain an average speed for each calcium wave. Next, this average speed was plotted (Figure 4B) and used to calculate a new average of all the calcium waves. The average velocity of calcium propagation was, at 3.1 µm/s (CaMV) and 3.3 µm/s (TuMV), significantly lower in the leaves infected with these viruses than in the healthy control leaves (4.0 µm/s). TuYV infection did not change the speed of calcium signals (3.8 µm/s) in comparison to that of the healthy control leaves.

### 3.3. CaMV Transmission from Arabidopsis Mutants Impaired in Pathogen Recognition or Signalling

CaMV and TuMV transmission by aphids from infected leaves or protoplasts is inhibited by lanthanum [15,16]. This suggests that virus acquisition requires calcium and could be triggered by the mechanical stress [29] caused by aphid punctures or by effector molecules contained in aphid saliva and injected during probing activity into the plant tissue [30]. Therefore, we tested the capability of several *Arabidopsis* mutants impaired in pathogen perception or of cationic channels to transmit CaMV. For this, we chose the central hub in plant innate immunity BAK1, which is involved in plant–aphid interactions [2,11] and some receptor-like kinases (RLK) involved in chitin and peptidoglycan perception, which could potentially participate in aphid perception. As cationic channels, TPC1 and several mechanosensitive channels were chosen. None of the mutants showed a significant difference in their transmission of CaMV (Table 1).

### 3.4. Screen for Genes Potentially Involved in Aphid Perception

Finally, we tested whether the reduced velocity of calcium waves in CaMV- and TuMV-infected leaves could be correlated with the modified expression of distinct *Arabidopsis* genes, as this might be evidence for a function of these genes in the perception of aphid punctures. For this, we screened publicly available transcriptomes from CaMV- and TuMV-infected *Arabidopsis* (see Section 2) for differentially expressed genes (DEGs) known to be involved in the perception of pathogens or in the generation of calcium signals (see Appendix A for a list of candidates). Since Vincent and co-workers [11] established an effect of the receptor-like kinase BAK1 in aphid recognition, we concentrated on genes located in the BAK1 pathway. Several differentially expressed *BAK1*-related genes were identified, but *BAK1* expression itself was not changed (Table 2). Curiously, all DEGs except one were different for the CaMV- and TuMV-infected plants, and the only gene in common for infection with either virus (*BIR1*) was upregulated in CaMV-infected and downregulated in TuMV-infected *Arabidopsis*. Concerning calcium-related genes, we found that the calcium channel *GLR2*.8 was upregulated during CaMV infection and downregulated in TuMV infection, whereas the channel *GLR2*.7 was downregulated in TuMV-infected and *GLR3*.7 in CaMV-infected plants. Of the cyclic nucleotide-gated channels, only *ATCNGC4* was upregulated and exclusively in the CaMV-infected plants, whereas four other CNCGs were downregulated in the TuMV-infected plants (Table 2). We also screened a transcriptome of TuYV-infected *Arabidopsis* for DEGs (Véronique Ziegler-Graff, personal communication). Using the standard threshold value of twofold change in expression, no gene from our candidate list was differentially expressed in TuYV-infected *Arabidopsis*.

## 4. Discussion

### 4.1. Comparison between Our Results and Those Obtained by Vincent et al.

Here we show that calcium signals could be monitored upon aphid punctures on detached leaves infected with viruses. Calcium signals triggered by aphid punctures in leaves have been shown before by Vincent and coworkers [11], but only in healthy tissues. The previously observed calcium signals [11] are different from the calcium excitations described here. The signals analysed in [11] are large, mostly linear, and long-lasting (~10 min), whereas we report here small ring-like calcium signals of short duration (<1 min). Thus, we analysed other waveforms than Vincent and co-workers did. There are several technical reasons to explain why we recorded different signals. The most likely is that Vincent et al. used a stereomicroscope with a low magnification (7.8×) to image calcium signals in the whole leaf, where the small excitations observed here might have passed unnoticed. We used higher magnification (50×) and focused on short, small, ring-like calcium signals and might have overseen the large, long-lasting excitations because their extent exceeded the observation field.

Vincent and co-workers used in parallel the electrical penetration graph technique to show that the promptness of the calcium signals was compatible with the onset of aphid probing on a leaf. Transferred to our work, this is strong evidence that the calcium elevations observed here are also caused by aphid activity. Other evidence that the ring-like calcium signals are induced by aphid punctures is the fact that they were found close to the aphids’ heads but not close to the back legs and also not when aphids moved (see Appendix A) and that we observed the annular calcium waves only on aphid-infested leaves. The ring signals were also not caused by wounding stress afflicted to the leaves when detaching them for observation, because at least 30 min passed between detachment and the start of recording. In fact, Vincent and co-workers used either whole plants or detached leaves for the experiments and did not find any differences in calcium responses, provided the leaves were allowed to recover from the wounding stress for a sufficient time. This is also similar to our own results published previously in [27], and indicates that the use of detached leaves, as was done throughout the present work, is a valid approach.

The same authors measured a velocity of 6 µm/s, which is faster than our measurements (4 µm/s average in healthy leaves, with directional velocity peaks of up to almost 5 µm/s). Two reasons may explain the differences. As mentioned above, the signals analysed by Vincent and co-workers probably represent another type of calcium signal then the ring-like excitations. Another reason is technical. We used a higher magnification (50×) and an inverted microscope to achieve better resolution, while the previous work used a binocular loupe with a lower image resolution (7.8×), so the accuracy of measurement was different. It is also noteworthy that the leaves used for imaging, especially the infected ones, were not flat but curled and the speed might have been underestimated. This, in addition to other differences in the experimental setup (calcium reporter, age of plants) could alternatively explain the different velocities.

Taken together, the different properties of the calcium signals reported by Vincent et al. and those presented here suggest that we analysed different calcium signals that are both related to aphid probing.

### 4.2. Calcium Propagation Is Slower in CaMV- and TuMV-Infected Leaves

CaMV and TuMV infection reduced the speed of localised calcium signals in systematically infected leaves by around 22.5% (from 4 µm/s in healthy to 3.1 µm/s and 3.3 µm/s in CaMV- and TuMV-infected leaves, respectively, Figure 4B). This coincided for CaMV, but not TuMV infection with slightly but significantly reduced signal areas (Figure 3B). It is known that *Myzus persicae* aphids find the phloem faster on CaMV-infected *Arabidopsis* than on healthy plants and need less punctures for this [31]. Whether and how the slower and smaller (for CaMV-infected plants) calcium signals contribute to this behaviour remains unknown. One could speculate that dampened aphid perception, as suggested by the weaker calcium elevations, coupled with a reduced immediate defence reaction at feeding sites, favoured faster stylet progression in the tissue, but further research is needed to resolve this question. TuYV infection did not have an effect on calcium velocity. TuYV is, unlike the tissue generalists CaMV and TuMV, a phloem-restricted virus. However, most aphid punctures were recorded in the interveinal tissues, which are not infected and probably unaffected by TuYV. It is plausible that, for this reason, no modified calcium signals were observed in the TuYV-infected plants. In a few instances, we observed in infected and healthy leaves calcium elevations triggered by aphids, which originated in the vascular system and propagated along the veins (Appendix A). Unfortunately, their number was too small to allow analysis, so it remains unknown whether infection with TuYV (or with CaMV and TuMV) alters vascular calcium signals.

### 4.3. Impact of Slower Calcium Propagation on Plant–Aphid Interactions

What might be the impact of dampened calcium elevations in CaMV- and TuMV-infected leaves? The work by Vincent et al. [11] showed that a mutation in *BAK1* (*bak1-5*), a key protein in pathogen recognition, led to lower aphid-triggered calcium elevations but that neither aphid feeding behaviour nor fecundity were modified [2]. These authors also showed that aphids elicited larger calcium elevations in the “overexcited” *TPC1* mutant *fou2*, which correlated with jasmonic acid-induced (JA) hormone responses and decreased aphid fitness. This suggests that BAK1-mediated calcium elevations induce JA responses that decrease aphid fecundity. In this context, the dampening of calcium signalling by CaMV and TuMV might favour aphid fitness by lowering defence responses on infected plants. However, the scenario might be more complicated and requires further investigation. Further, the role of hormones could be virus-dependent. For example, CaMV infection decreases aphid fecundity on infected *Arabidopsis* [31], and this correlates with increased JA and ethylene (ETH) signalling responses [32,33]. On the other hand, increased fecundity was reported for aphids on TuMV-infected *Arabidopsis* and correlated with higher ethylene levels [34]. At the same time, aphid-induced JA accumulation was not suppressed by viral infection. This indicates that other pathways contribute to plant–virus–aphid interactions as well, and that the precise molecular links between diminished calcium signalling and the different defence responses remain to be established. There are no data on the hormone responses of *Arabidopsis* during TuYV infection, but the related polerovirus potato leafroll (PLRV) increases aphid fecundity on potato, and this correlates with the suppression of aphid-induced JA/ETH defences in PLRV-infected plants [35]. However, the possible downregulation of JA/ETH defences in TuYV-infected plants seems not to be related to slower, aphid-triggered, ring-like calcium elevations, since they were not modified by TuYV infection.

### 4.4. Role of BAK1 in Calcium-Mediated Aphid–Plant Interactions

BAK1 is a central co-receptor in pathogen recognition. It interacts with other receptor-like kinases (RLKs) to form functional signalling complexes. We screened transcriptomic data (see Section 2 and Appendix A) for genes in the *BAK1* pathway and cation channel genes that are differently expressed in CaMV-, TuMV-, and TuYV-infected versus healthy *Arabidopsis* and that could thus be involved in aphid–plant interactions. We found several genes that were deregulated by CaMV or TuMV infection, but no differentially expressed gene (DEG) in TuYV-infected *Arabidopsis*, using a more than twofold expression change as the selection criterion for the minimum significance threshold. Therefore, TuYV does not seem to significantly modify the expression of cation- and *BAK1*-related genes. Table 2 shows that *BAK1* expression was not significantly different in CaMV- and TuMV-infected plants compared to healthy plants. However, the expression of several co-receptor kinases and BAK1-interacting proteins was modified, four (two up- and two downregulated) in CaMV and five (all downregulated) in TuMV-infected *Arabidopsis*. Only one gene (*BIR1*) was common for both infections. Interestingly, *BIR1* was upregulated in CaMV- and downregulated in TuMV-infected plants, and has previously been implied to contribute to antiviral defence [36]. How and whether *BIR1* and other DEGs in the BAK1 pathway (Table 2) function in aphid perception remains to be determined. We propose that BAK1, in its function as a general information-transducing hub, is indeed involved in plant aphid defences, and that BAK1-interacting RLKs confer specificity to the aphid defence response and activate the BAK1 pathway.

### 4.5. Involvement of Different Ion Channels in Aphid-Triggered Calcium Elevations

The analysis of the expression of cation channels showed that none of them were deregulated in TuYV-infected *Arabidopsis*. This reflected the finding that TuYV infection did not alter calcium elevations. Several channel genes were differentially expressed in TuMV- and CaMV-infected plants. The expression of the vacuolar transporter *TPC1* and the plasma membrane channels *GLR3.3* and *GLR3.6*, which were shown by Vincent et al. [11] to contribute to aphid-induced calcium propagation, was not significantly affected or was less than twofold (*GLR3.3* in TuMV infection). This indicates that the dampening of calcium elevations by the two viruses is achieved by interaction with other calcium channels or is due to post-translational modifications. The expression of two other GLRs, plasma membrane- and/or tonoplast-localised GLR2.7 and plasma membrane-localised GLR2.8, was downregulated in TuMV-infected plants. Curiously, *GLR2.8* transcripts were significantly upregulated during CaMV infection. Finally, the plasma membrane channel GLR3.7 was significantly downregulated in CaMV, but not in TuMV. We also looked at the expression of cyclic nucleotide-gated calcium channels (CNGCs). One *CNGC* gene, *CNGC4*, was upregulated only in CaMV-infected *Arabidopsis*, whereas the expression of four other channels (*CNGC3*, *10*, *14*, and *19*) was exclusively downregulated in TuMV-infected plants. One of these channels, CNGC19, is involved in defences against herbivores such as *Spodoptera* caterpillars, and *CNGC19* loss of function mutants are more susceptible to herbivores [37]. Whether CNGC19 downregulation impacts aphid–plant interactions is unknown. It is usually assumed that herbivory and aphid feeding induces different plant hormone-mediated defence pathways, although it has not been ruled out that they overlap partially [38,39].

CaMV and TuMV acquisition by aphid vectors is very rapid and is inhibited by the calcium channel blocker lanthanum [15,16]. Therefore, we speculated that calcium channel mutants or mutants affected in the BAK1 pathway might interfere with virus transmission. Using infected mutant plants as the virus source, no difference in the transmission efficiency of CaMV was observed (Table 1). Since the mutant *bak1-5* was not impaired in CaMV transmission, the most plausible explanation is that the BAK1-mediated defence response pathways are not needed for transmission or that local effects in different tissues and cell types masked the influence of the pathways. This might also explain why none of the tested *BAK1*-related *RLK* mutants showed a significant difference in their transmission of CaMV when used as the virus source. Also, *Arabidopsis* mutants affected in various cation channels did not show any altered CaMV transmission. This suggests that the implied channels are not involved in transmission or that functional redundancy masked any effect. Alternatively, CaMV infection inactivated these channels independently of vector contact, explaining the slower calcium signals and the missing effect on transmission. We used non-optimal conditions for the transmission experiments with the *tpc1.2*, *fou2*, and the quintuple mechanosensitive channel knockout mutant *M5* [40], i.e., rather long acquisition times (15 min) and two aphids per test plant. This might have promoted transmission and possibly biased the results compared to transmission tests using shorter virus acquisition times and only one aphid per test plant. Having said that, an interesting hypothesis is that the rapid calcium-dependent acquisition of CaMV and TuMV requires the fast, small, ring-like calcium excitations reported here, while the longer-lasting calcium signals reported by Vincent et al. do not relate to transmission, but to plant defences. One could also hypothesise that the short annular calcium waves analysed here prime the generation of the large and long-lasting calcium signals and that only the latter signals trigger defence responses.

Taken together, a complex picture emerges where aphid perception might be achieved by the specific cooperation of distinct calcium channels and different RLKs. CaMV and TuMV, but not TuYV, seem to manipulate this recognition system virus-specifically by deregulating distinct calcium channels and RLKs. For TuYV, this might mean that this virus uses a “stealthy lifestyle” and impacts the host’s physiology, including signalling pathways, as little as possible. This also manifests itself by the very weak symptoms in TuYV-infected *Arabidopsis* [41]. For CaMV and TuMV, this indicates that these two viruses modify aphid/plant interactions from the very first physical encounter between the two to help “hide” aphids from the plant host and favour the virus–aphid interactions ultimately required for transmission. Whether calcium-dependent virus acquisition uses the same pathways remains unclear. The viral modulation of calcium responses might also modify interactions other than aphid–plant interactions. The future perspectives of this work will be focused on the identification of mechanisms targeted by the viruses.

## Figures and Tables

**Figure 1 cells-10-03534-f001:**
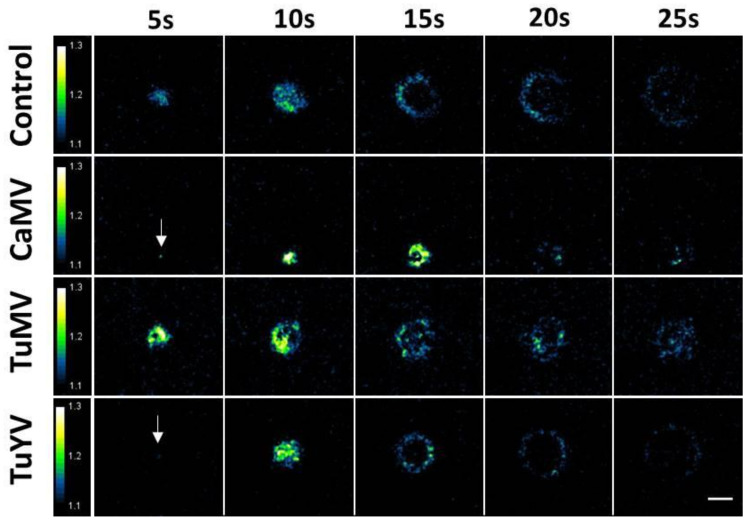
Aphid punctures trigger calcium signals in *Arabidopsis* leaves. An aphid was placed on the upper side of a detached leaf of transgenic *Arabidopsis* plants expressing the calcium reporter protein YC3.6-NES. Images of calcium elevations were captured every five seconds by ratiometric epifluorescence microscopy. The figure shows time series of calcium waves induced by aphid punctures in healthy control leaves or in leaves infected with the indicated virus. The arrows in the CaMV and TuYV time series point to the origin of the excitation. The relative signal intensity is shown in false colours. The scale bar is 0.1 mm.

**Figure 2 cells-10-03534-f002:**
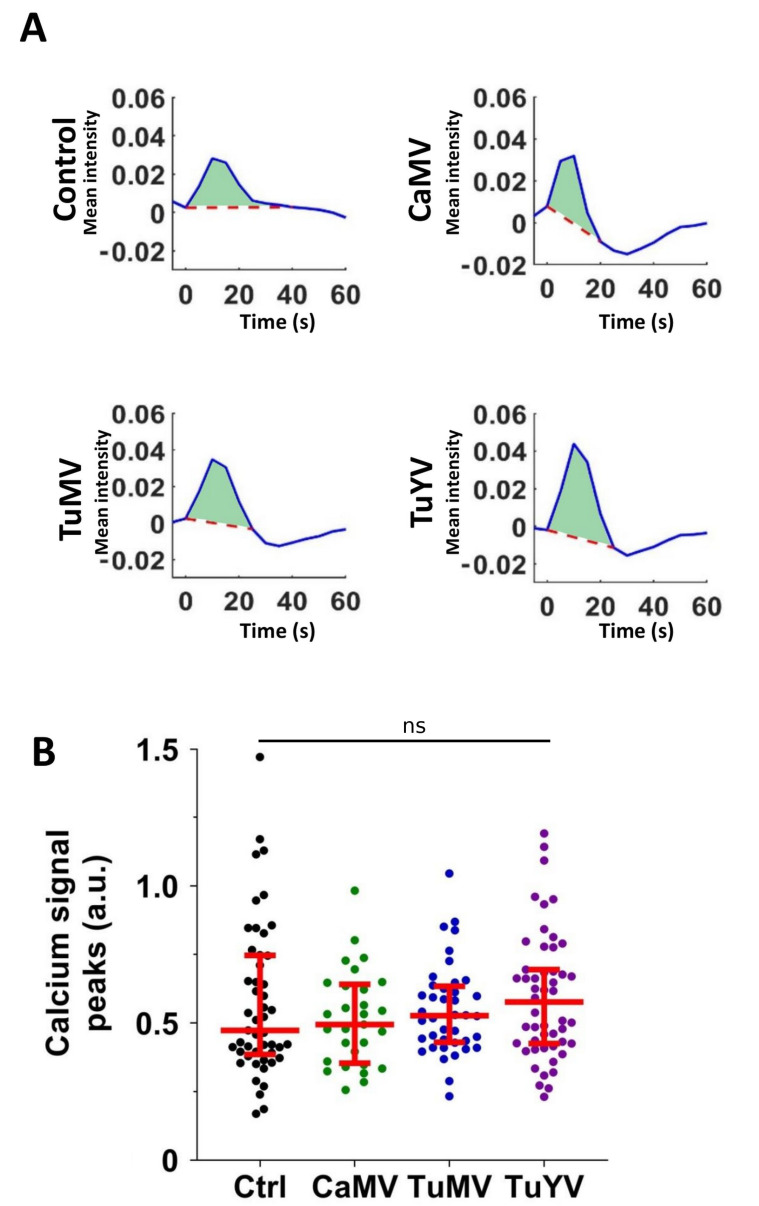
Quantification of the calcium signal peaks. (**A**) Representative plots of time series of normalized calcium signal mean intensity of derivative images. The light green areas indicate the calcium signal peaks. (**B**) All signal peaks (closed circles) in arbitrary units (a.u.) for each condition (healthy control (Ctrl) or infection with the indicated virus). The horizontal red lines indicate medians and the 1st and 3rd quartiles, respectively. ns = Non-significant (one-way ANOVA test).

**Figure 3 cells-10-03534-f003:**
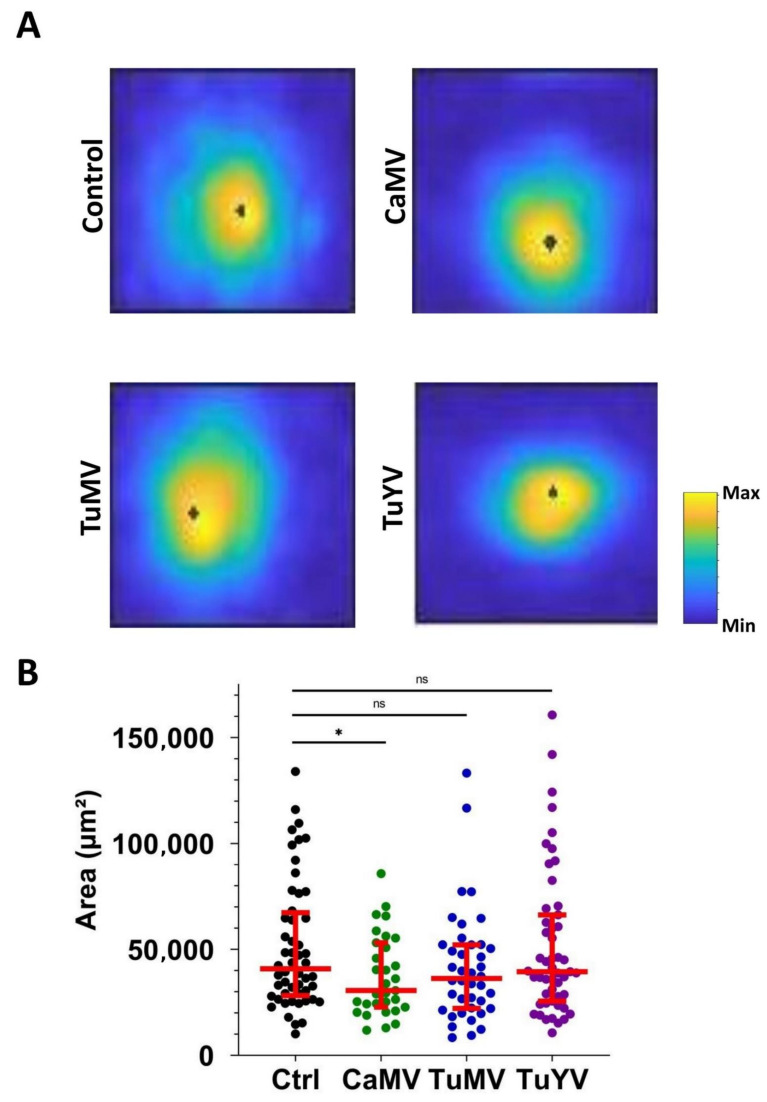
Analysis of calcium signal areas. (**A**) Representative projections of maxima of aphid-triggered calcium signals for each condition (healthy control (Ctrl) or infected with the indicated virus). The scale indicates the calcium level in arbitrary units and the black dots in the circular signals indicate the point of origin of the calcium signal. (**B**) Quantification of the calcium signal areas. Each dot presents the signal area of a calcium elevation triggered by an aphid puncture. The conditions (healthy control (Ctrl) or virus infection) are indicated below the graph. The horizontal red lines indicate the medians and the 1st and 3rd quartiles, respectively. Mann–Whitney test was performed. Significant results were found. * *p*-value = 0.0383, ns = non-significant.

**Figure 4 cells-10-03534-f004:**
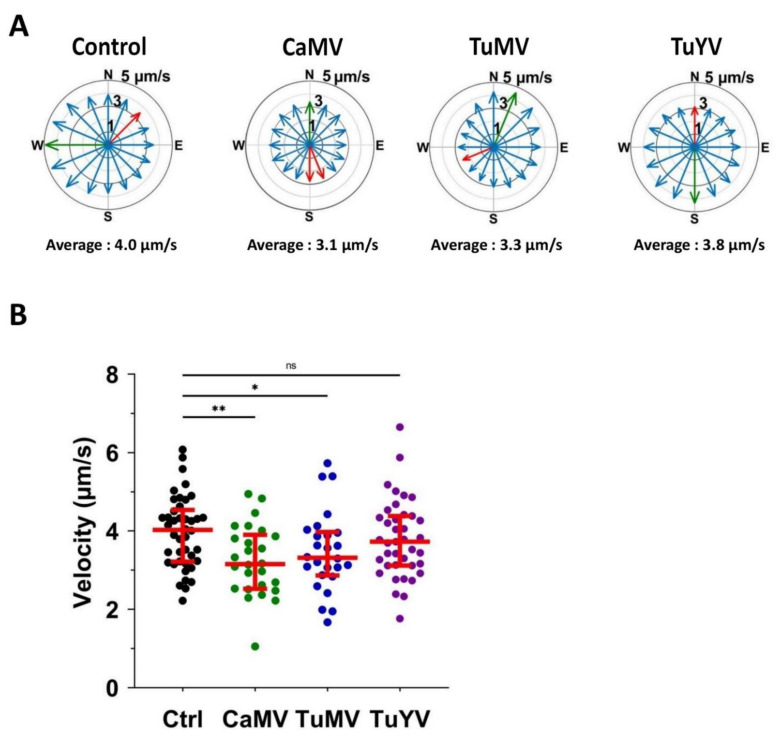
Direction and velocity of aphid-induced calcium waves. (**A**) The plots show the angle-dependent velocity for a representative example of each condition (healthy control (Ctrl) or infection with the indicated virus). The position of the arrows presents the angle, the length of the arrows the velocity, while the red and green arrows indicate the lowest and highest speed of the signal, respectively. The average speed of the shown calcium wave is indicated below each graph. (**B**) Quantification of the velocity. The mean velocity of each calcium signal is plotted for all healthy control samples (Ctrl) and for samples infected with CaMV, TuMV, and TuYV. The horizontal red lines indicate medians and the 1st and 3rd quartiles, respectively. Significant results were found. *p*-value ** = 0.0021, * = 0.0397, ns = non-significant (*t*-test).

**Table 1 cells-10-03534-t001:** Transmission tests using CaMV-infected Col-0 or the indicated *Arabidopsis* mutant plants as the virus source. Aphids were allowed to acquire the virus from the indicated infected source plants and then were transferred to healthy turnip seedlings for inoculation. Symptoms were scored three weeks later by visual inspection. Tests where no transmission was observed in one condition were not taken into account. Statistical analysis was by Fisher’s exact test. Acquisition time was 1 min except for *bak1-5* (15 s) and *tpc1.2*, *fou2*, and *M5* (15 min). One aphid was transferred for each transmission test except for *tpc1.2*, *fou2*, and *M5*, where two aphids were used for inoculation. * SD, standard deviation of transmission tests; ^#^ WT, wild type.

		% Transmission ± SD * Using as Source			Plants Inoculated			
TAIR Locus	Mutant	Mutant	WT ^#^	*n*	Mutant/WT	*p*-Value	Seeds from	References
**Signal transduction**								
AT4G33430	*bak1-5*	45.0 ± 13.9	52.0 ± 18.2	7	138/130	0.39	NASC N799997	https://doi.org/10.1371/journal.pgen.1002046 (accessed on 4 March 2021)
AT3G21630	*lyk1=CERK1*	36.2 ± 14.8	37.1 ± 15.9	12	221/240	0.85	GABI-KAT 096F09	https://doi.org/10.1073/pnas.0705147104 (accessed on 4 March 2021)
AT3G01840 AT1G51940 AT2G33580	*lyk2 lyk3 lyk5*	48.3 ± 12.8	47.5 ± 12.0	11	214/217	0.85	Gary Stacey	https://doi.org/10.1104/pp.112.201699 (accessed on 4 March 2021)
AT2G23770	*lyk4*	50.4 ± 9.8	54.2 ± 14.7	12	233/227	0.46	Gary Stacey	https://doi.org/10.1104/pp.112.201699 (accessed on 4 March 2021)
AT1G77630	*lyp3*	32.6 ± 16.3	38.6 ± 19.1	11	216/220	0.19	NASC SALK_132566	https://doi.org/10.1073/pnas.1112862108 (accessed on 4 March 2021)
**Calcium signal regulation**								
AT4G03560	*tpc1.2*	53.7 ± 14.2	50.2 ± 6.6	5	188/199	0.10	Saskia Hogenhout	https://doi.org/10.1038/nature03381 (accessed on 4 March 2021)
AT4G03560	*fou2*	46.9 ± 11.5	36.6 ± 7.9	4	96/93	0.19	Edward Farmer	https://doi.org/10.1111/j.1365-313x.2006.03002.x (accessed on 4 March 2021)
AT1G53470 AT3G14810 AT1G78610 AT5G19520 AT5G12080	*M5*	42.7 ± 27.2	47.2 ± 29.2	14	334/330	0.28	Dominique Roby	https://doi.org/10.1016/j.cub.2008.04.039 (accessed on 4 March 2021)
AT4G35920 AT2G17780	*mca1 mca2*	52.9 ± 13.9	50.9 ± 14.1	17	326/329	0.76	Hidetoshi Iida	https://doi.org/10.1104/pp.109.147371 (accessed on 4 March 2021)

**Table 2 cells-10-03534-t002:** *BAK1*-related and calcium signal-related genes differentially expressed in CaMV- and TuMV-infected *Arabidopsis*. Values highlighted in green or red correspond to significantly deregulated genes (*p*-value < 0.05 and more than twofold changed expression compared to mock-infected controls). See Appendix A for a list of all genes tested. For comparison, expression values for *BAK1*, *TPC1*, *GLR3.3*, and *GLR3.6* that have been implicated in plant responses to aphids [11] are shown. All values are in log2-fold change of expression (log2FC).

TAIR Locus ID	Name	CaMV (log2FC)	TuMV (log2FC)
**Signal transduction regulator**			
AT4G33430	*BAK1*	−0.084	−0.236
AT5G48380	*BIR1*	2.764	−1.145
AT4G32910	*SBB1*	1.414	<1
AT5G44585	*PROSCOOP12*	<1	−1.027
AT2G17120	*LYM2*	−1.099	<1
AT5G66210	*CPK28*	<1	−1.204
AT3G21630	*LYK1*	−1.296	<1
AT2G13790	*SERK4*	<1	−1.722
AT1G51850	*SIF2*	<1	−2.394
		<1	<1
**Calcium signal generation**			
AT4G03560	*TPC1*	−0.126	0.231
AT1G42540	*GLR3.3*	−0.079	0.442
AT3G51480	*GLR3.6*	−0.381	0.268
AT2G29110	*GLR2.8*	1.074	−2.608
AT2G29120	*GLR2.7*	<1	−1.136
AT2G32400	*GLR3.7*	−1.616	<1
AT5G54250	*ATCNGC4/ATDND2/ATHLM1*	1.048	<1
AT2G46430	*ATCNGC3*	<1	−1.025
AT2G24610	*ATCNGC14*	<1	−1.317
AT1G01340	*ATCNGC10/ATACBK1*	<1	−1.339
AT3G17690	*ATCNGC19/ATCNBT2*	<1	−2.266

## Data Availability

The data presented in this study are available on request from the corresponding authors.

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
