# Peer review of "Plant Viruses Can Alter Aphid-Triggered Calcium Elevations in Infected Leaves"

_cells, 2021, doi:10.3390/cells10123534_

Round 1
Reviewer 1 Report
The manuscript by Then et al describe a detection the calcium signal during the infection by plant viruses by using a transgenic Arabidopsis line expressing the cytosolic calcium reporter YC3.6-NES.
Generally, the results are interesting, while the result section is lacking the subtitle, hence it is not easy to follow what the authors want to express.
1 For lines 193-195, the interpretation of Figure 1 is too simple, and should be more detail
2 Why the acquisition time was different in different mutants (i.e. bak1-5 (15 s) and tpc1.2, fou2 and M5 (15 min), and other mutants for 1 min)?.
3 In line 307, it is better to cite the Vincent et al. (2017) as the digital number
4 For the names of treatments, sometimes they are Italics, while sometimes are not.
5 In Figure 3B and 4B, what the ctrl presents?
6 For Figure 2B, the significant analysis should be performed.
7 SD* in the table 2 is not consistent with line 275
8 Why did the author using short day-light condition in the transgenic Arabidopsis line?
Author Response
Response to Reviewer 1
Reviewer 1
The manuscript by Then et al describe a detection the calcium signal during the infection by plant viruses by using a transgenic Arabidopsis line expressing the cytosolic calcium reporter YC3.6-NES.
Generally, the results are interesting, while the result section is lacking the subtitle, hence it is not easy to follow what the authors want to express.
Our answer: This is a nice idea. We added subtitles to the results sections.
1 For lines 193-195, the interpretation of Figure 1 is too simple, and should be more detail
Our answer: Interpretation of Figure 1 is now more detailed in the text. Please see the track changes in the revised manuscript.
2 Why the acquisition time was different in different mutants (i.e. bak1-5 (15 s) and tpc1.2, fou2 and M5 (15 min), and other mutants for 1 min)?
Our answer: The experiments were done at different time points by different people. Other labs employ still different aphid numbers and acquisition times for CaMV transmission experiments. Therefore, we think that the outcome of the experiments was not significantly biased by the different acquisition conditions. However, we agree, that it would have been better to do all transmission assays using identical conditions.
3 In line 307, it is better to cite the Vincent et al. (2017) as the digital number
Our answer: The Vincent citation is now in brackets “[11]” where appropriate.
4 For the names of treatments, sometimes they are Italics, while sometimes are not.
Our answer: We standardized the format of the names of the treatments in all figures. They are now all in normal style.
5 In Figure 3B and 4B, what the ctrl presents?
Our answer: Ctrl (healthy control) is now explained throughout the figure legends.
6 For Figure 2B, the significant analysis should be performed.
Our answer: We added the significance analysis (one-way ANOVA) to the figure legend and inserted a bar and ‘ns’ (not significant) into figure 2B.
7 SD* in the table 2 is not consistent with line 275
Our answer: “*SD, standard deviation” refers to SD in “% transmission ± SD” in the 3rd column of Table 2. We added “standard deviation of transmission tests” to the legend for more clarity.
8 Why did the author using short day-light condition in the transgenic Arabidopsis line?
Our answer: We used short day conditions to slow plant growth and infection progress to allow for a greater time window to work with plants from the same lot.
Reviewer 2 Report
The manuscript by Then and coauthors reports an original research effort to unveil the putative alteration of calcium signalling in plants responding to aphid probing, caused by the presence of plant viruses. The work is done with Arabidopsis plants and includes three different viruses, two of them following a non-persistent mode of transmission, and the third one being a persistent virus, allowing to correlate their observations with the modes of transmission. By using a sophisticated methodology the authors succeded to monitor the calcium responses in plants after aphid puncture and measure timings, peaks, and area of spread of the responses, including directions and velocity. The results of these parts are rather complete, and all together represents a remarkable descriptive effort, with many novel observations. Elaborating on that, authors select available mutants of genes described to regulate perception of pathogens and signalling to perform transmission experiments. When tested for virus acquisition, no effects on transmission efficiencies were observed for any of the 9 mutants. Also, a meta-analysis (using publicly available data) served to identify differentially expressed genes in virus infected plants: interestingly, no gene of the list of candidates was found for the persitently transmitted virus. It is also worth to highlight that some genes showed opposite responses (up and down) depending on the virus, probably reflecting the expected variability among different viruses, rather different taxonomically: a caulimovirus and a potyvirus, with dsDNA or ssRNA genomes
The paper is well written, easy to follow and with adequate figures and tables to illustrate the results. The experiments were very well designed, and it is remarkable the large size of samples used for most of the experiments, which contributes to the general robustness of the results. However, the discussion is too long (> 3 pages), and it might be shortened. In particular the first section is devoted to compare results with a previous work by other team: in my opinion there is no need to discuss so extensively every single difference.
Suggestions for modifications/additions:
- P. 3, l. 124. Do aphids change the position of their antenna when feeding? Was this behavior considered for measuring times?
- Reduce the discussion section, in particular the first section
- P. 13, first paragraph, it would be interesting to consider the different "lifestyles" of CaMV and TuMV as another alternative to explain the data.
- P. 13, l. 446-450. It could de adequate to consider here the possibility that local effects in different tissues and cell types might be masking the influence of the pathways
Author Response
Response to Reviewer 2
Reviewer 2
The manuscript by Then and coauthors reports an original research effort to unveil the putative alteration of calcium signalling in plants responding to aphid probing, caused by the presence of plant viruses. The work is done with Arabidopsis plants and includes three different viruses, two of them following a non-persistent mode of transmission, and the third one being a persistent virus, allowing to correlate their observations with the modes of transmission. By using a sophisticated methodology the authors succeded to monitor the calcium responses in plants after aphid puncture and measure timings, peaks, and area of spread of the responses, including directions and velocity. The results of these parts are rather complete, and all together represents a remarkable descriptive effort, with many novel observations. Elaborating on that, authors select available mutants of genes described to regulate perception of pathogens and signalling to perform transmission experiments. When tested for virus acquisition, no effects on transmission efficiencies were observed for any of the 9 mutants. Also, a meta-analysis (using publicly available data) served to identify differentially expressed genes in virus infected plants: interestingly, no gene of the list of candidates was found for the persitently transmitted virus. It is also worth to highlight that some genes showed opposite responses (up and down) depending on the virus, probably reflecting the expected variability among different viruses, rather different taxonomically: a caulimovirus and a potyvirus, with dsDNA or ssRNA genomes
The paper is well written, easy to follow and with adequate figures and tables to illustrate the results. The experiments were very well designed, and it is remarkable the large size of samples used for most of the experiments, which contributes to the general robustness of the results. However, the discussion is too long (> 3 pages), and it might be shortened. In particular the first section is devoted to compare results with a previous work by other team: in my opinion there is no need to discuss so extensively every single difference.
Our answer: Thank you for the comments. We shortened the first part of the discussion. Please refer to the track changes in the revised manuscript.
Suggestions for modifications/additions:
- P. 3, l. 124. Do aphids change the position of their antenna when feeding? Was this behavior considered for measuring times?
Our answer: They lay them on their back when feeding. We added to the text “…when the aphid stopped moving and positioned the antennae on its back, which is an indication that it had started feeding behaviour.”
- Reduce the discussion section, in particular the first section
Our answer: Please see our comment above.
- P. 13, first paragraph, it would be interesting to consider the different "lifestyles" of CaMV and TuMV as another alternative to explain the data.
Our answer: This is an excellent idea. We added a corresponding sentence to the concluding paragraph of the manuscript: “For TuYV, this might mean that this virus uses a “stealthy lifestyle” and impacts the host’s physiology including signalling pathways the least possible. This manifests itself also by the very weak symptoms in TuYV-infected Arabidopsis [42]. For CaMV and TuMV, this indicates that these two viruses modify aphid/plant interactions from the very first physical encounter between the two to help “hide” aphids from the plant host and favour virus-aphid interactions ultimately required for transmission.”
- P. 13, l. 446-450. It could de adequate to consider here the possibility that local effects in different tissues and cell types might be masking the influence of the pathways
Our answer: Thanks for this suggestion. We added the sentence “Since the mutant bak1-5 was not impaired in CaMV transmission, the most plausible explanation is that the BAK1-mediated defence response pathways are not needed for transmission or that local effects in different tissues and cell types masked the influence of the pathways.”